# Meta-Reinforced Synthetic Data for One-Shot Fine-Grained Visual Recognition

**Satoshi Tsutsui**
Indiana University
USA
stsutsui@indiana.edu

**Yanwei Fu**[*]
Fudan University
China
yanweifu@fudan.edu.cn

**David Crandall**
Indiana University
USA
djcran@indiana.edu

## Abstract

One-shot fine-grained visual recognition often suffers from the problem of training data scarcity for new fine-grained classes. To alleviate this problem, an off-the-shelf image generator can be applied to synthesize additional training images, but these synthesized images are often not helpful for actually improving the accuracy of one-shot fine-grained recognition. This paper proposes a meta-learning framework to combine generated images with original images, so that the resulting "hybrid" training images can improve one-shot learning. Specifically, the generic image generator is updated by a few training instances of novel classes, and a Meta Image Reinforcing Network (MetaIRNet) is proposed to conduct one-shot fine-grained recognition as well as image reinforcement. The model is trained in an end-to-end manner, and our experiments demonstrate consistent improvement over baselines on one-shot fine-grained image classification benchmarks.

## 1 Introduction

The availability of vast labeled datasets has been crucial for the recent success of deep learning. However, there will always be learning tasks for which labeled data is sparse. Fine-grained visual recognition is one typical example: when images are to be classified into many very specific categories (such as species of birds), it may be difficult to obtain training examples for rare classes, and producing the ground truth labels may require significant expertise (e.g., ornithologists). One-shot learning is thus very desirable for fine-grained visual recognition.

A recent approach to address data scarcity is meta-learning [7, 10, 24, 35], which trains a parameterized function called a meta-learner that maps labeled training sets to classifiers. The meta-learner is trained by sampling small training and test sets from a large dataset of a base class. Such a meta-learned model can be adapted to recognize novel categories with a single training instance per class. Another way to address data scarcity is to synthesize additional training examples, for example by using off-the-shelf Generative Adversarial Networks (GANs) [3, 13]. However, classifiers trained from GAN-generated images are typically inferior to those trained with real images, possibly because the distribution of generated images may be biased towards frequent patterns (modes) of the original image distribution [26]. This is especially true in one-shot fine-grained recognition where a tiny difference (e.g., beak of a bird) can make a large difference in class.

To address these issues, we develop an approach to apply off-the-shelf generative models to synthesize training data in a way that improves one-shot fine-grained classifiers. We begin by conducting a pilot study to transfer a generator pre-trained on ImageNet in a one-shot scenario. We show that the generated images can indeed improve the performance of a one-shot classifier when used with

---

[*]Y. Fu is with School of Data Science, and MOE Frontiers Center for Brain Science, Shanghai Key Lab of Intelligent Information Processing, Fudan University.

a carefully-designed rule to combine the generated images with the originals. Based on these preliminary results, we propose a meta-learning approach to learn these rules to reinforce the generated images effectively for few-shot classification.

Our approach has two steps. First, an off-the-shelf generator trained from ImageNet is updated towards the domain of novel classes by using only a single image (Sec. 4.1). Second, since previous work and our pilot study (Sec. 3) suggest that simply adding synthesized images to the training data may not improve one-shot learning, the synthesized images are "mixed" with the original images in order to bridge the domain gap between the two (Sec. 4.2). The effective mixing strategy is learned by a meta-learner, which essentially boosts the performance of fine-grained categorization with a single training instance per class. Lastly, we experimentally validate that our approach can achieve improved performance over baselines on fine-grained classification datasets in one-shot situations (Sec. 5).

To summarize, the contributions of this paper are: (1) a method to transfer a pre-trained generator with a single image, (2) a method to learn to complement real images with synthetic images in a way that benefits one-shot classifiers, and (3) to experimentally demonstrate that these methods improve one-shot classification accuracy on fine-grained visual recognition benchmarks.

## 2   Related Work

**Image Generation.** Learning to generate realistic images has many potential applications, but is challenging with typical supervised learning. Supervised learning minimizes a loss function between the predicted output and the desired output but, for image generation, it is not easy to design such a perceptually-meaningful loss between images. Generative Adversarial Networks (GANs) [13] address this issue by learning not only a generator but also a loss function — the discriminator — that helps the generator to synthesize images indistinguishable from real ones. This adversarial learning is intuitive but is known to often be unstable [14] in practice. Recent progress includes better CNN architectures [3, 21], training stabilization tips [2, 14, 19], and interesting applications (e.g. [38]). In particular, BigGAN [3] trained on ImageNet has shown visually impressive generated images with stable performance on generic image generation tasks. Several studies [20, 33] have explored generating images from few examples, but their focus has not been on one shot classification. Several papers [8, 9, 20] use the idea of adjusting batch normalization layers, which helped inspire our work. Finally, some work has investigated using GANs to help image classification [1, 12, 26, 27, 37]; our work differs in that we apply an off-the-shelf generator pre-trained from a large and generic dataset.

**Few-shot Meta-learning.** Few shot classification [4] is a sub-field of meta-learning (or "learning-to-learn") problems, in which the task is to train a classifier with only a few examples per class. Unlike the typical classification setup, in few-shot classification the labels in the training and test sets have no overlapping categories. Moreover, the model is trained and evaluated by sampling many few-shot tasks (or episodes). For example, when training a dog breed classifier, an episode might train to recognize five dog species with only a single training image per class — a 5-way-1-shot setting. A meta-learning method trains a meta-model by sampling many episodes from training classes and is evaluated by sampling many episodes from other unseen classes. With this episodic training, we can choose several possible approaches to learn to learn. For example, "learning to compare" methods learn a metric space (e.g., [28, 29, 31]), while other approaches learn to fine-tune (e.g., [10, 11, 22, 23]) or learn to augment data (e.g., [6, 12, 15, 25, 34]). An advantage of the latter type is that, since it is data augmentation, we can use it in combination with any other approaches. Our approach also explores data augmentation by mixing the original images with synthesized images produced by a fine-tuned generator, but we find that the naive approach of simply adding GAN generated images to the training dataset does not improve performance. But by carefully combining generated images with the original images, we find that we can effectively synthesize examples that contribute to increasing the performance. Thus meta-learning is employed to learn the proper combination strategy.

## 3   Pilot Study

To explain how we arrived at our approach, we describe some initial experimentation which motivated the development of our methods.

Table 1: CUB 5-way-1-shot classification accuracy (%) using ImageNet features. Simply adding generated images to the training set does not help, but adding hybrid images, as in Fig. 1 (h), can.

| Training Data | Nearest Neighbor | Logistic Regression | Softmax Regression |
|---|---|---|---|
| Original | 69.6 | 75.0 | 74.1 |
| Original + Generated | 70.1 | 74.6 | 73.8 |
| Original + Mixed | 70.6 | 75.5 | 74.8 |

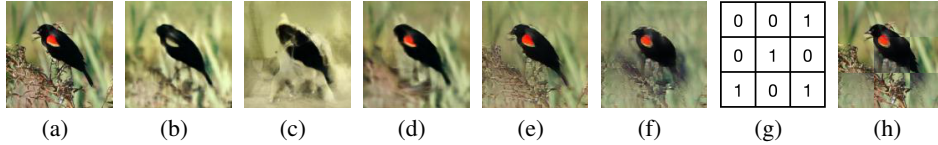

| (a) | (b) | (c) | (d) | (e) | (f) | (g) | (h) |

Figure 1: Samples described in Sec. 3. (a) Original image. (b) Result of tuning noise only. (c) Result of tuning the whole network. (d) Result of tuning batch norm only. (e) Result of tuning batch norm with perceptual loss. (f) Result of slightly disturbing noise from (e). (g) a $3 \times 3$ block weight matrix $w$. (g) Result of mixing (a) and (f) as $w \times$(f) $+ (1 - w) \times$(a).

**How can we transfer generative knowledge from pre-trained GANs?** We aim to quickly generate training images for few-shot classification. Performing adversarial learning (*i.e.* training generator and discriminator initializing with pre-trained weights) is not practical when we only have one or two examples per class. Instead, we want to develop a method that does not depend on the number of images at all; in fact, we consider the extreme case where only a single image is available, and want to generate variants of the image using a pre-trained GAN. We tried fixing the generator weights and optimizing the noise so that it generates the target image, under the assumption that sightly modifying the optimized noise would produce a variant of the original. However, naively implementing this idea with BigGAN did not reconstruct the image well, as shown in the sample in Fig. 1(b). We then tried fine-tuning the generator weights also, but this produced even worse images stuck in a local minima, as shown in Fig. 1(c).

We speculate that the best approach may be somewhere between the two extremes of tuning noise only and tuning both noise and weights. Inspired by previous work [8,9,20], we propose to fine-tune only scale and shift parameters in the batch normalization layers. This strategy produces better images as shown in Fig. 1(d). Finally, again inspired by previous work [20], we not only minimize the pixel-level distance but also the distance of a pre-trained CNN representation (i.e. perceptual loss [17]), and we show the slightly improved results in Fig. 1(e). We can also generate slightly different versions by adding random perturbations to the tuned noise (e.g., the "fattened" version of the same bird in Fig. 1(f)). The entire training process needs fewer than 500 iterations and takes less than 20 seconds on an NVidia Titan Xp GPU. We explain the resulting generation strategy developed based on this pilot study in Sec. 4.

**Are generated images helpful for few shot learning?** Our goal is not to generate images, but to augment the training data for few shot learning. A naive way to do this is to apply the above generation technique for each training image, in order to double the training set. We tested this idea on a validation set (split the same as [4]) from the Caltech-UCSD bird dataset [32] and computed average accuracy on 100 episodes of 5-way-1-shot classification. We used pre-trained ImageNet features from ResNet18 [16] with nearest neighbor, one-vs-all logistic regression, and softmax regression (or multi-class logistic regression). As shown in Table 1, the accuracy actually drops for two of the three classifiers when we double the size of our training set by generating synthetic training images, suggesting that the generated images are harmful for training classifiers.

**What is the proper way of synthesizing images to help few-shot learning?** Given that the synthetic images *appear* meaningful to humans, we conjecture that they can benefit few shot classification when properly mixed with originals to create hybrid images. To empirically test this hypothesis, we devised a random $3 \times 3$ grid to combine the images. As shown in Fig. 1(h), images (a) and (f) were combined by taking a linear combination within each cell of the grid of (g). Finally, we added mixed images like (h) into the training data, and discovered that this produced a modest increase in accuracy (last row of Table 1). While the increase is marginal, these mixing weights were binary and manually

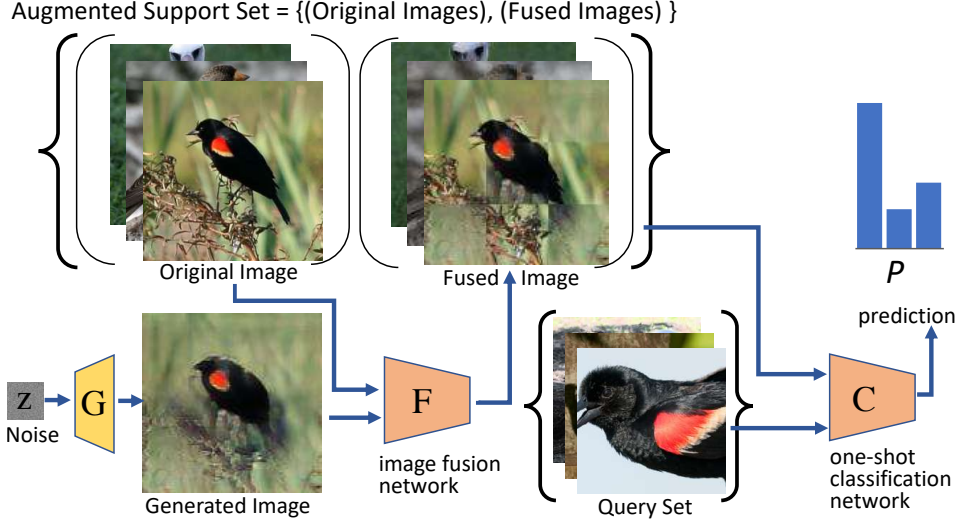

Augmented Support Set = {(Original Images), (Fused Images) }

Original Image  Fused Image  $P$
prediction

$z$ Noise  $G$  $F$  image fusion network  Query Set  $C$  one-shot classification network

Generated Image

Figure 2: **Our Meta Image Reinforcing Network (MetaIRNet)** has two modules: an image fusion network, and a one-shot classification network. The image fusion network reinforces generated images to try to make them beneficial for the one-shot classifier, while the one-shot classifier learns representations that are suitable to classify unseen examples with few examples. Both networks are trained end-to-end, so the loss back-propagates from classifier to the fusion network.

selected, and thus likely not optimal. In Sec. 4.2, we show how to learn this mixing strategy in an end-to-end manner using a meta-learning framework.

## 4   Method

The results of the pilot study in the last section suggested that producing synthetic images could be useful for few-shot fine-grained recognition, but only if it is done in a careful way. In this section, we use these findings to propose a novel technique for doing this effectively. We propose a GAN fine-tuning method that works with a single image (Sec. 4.1), and an effective meta-learning method to not only learn to classify with few examples, but also to learn to effectively reinforce the generated images (Sec. 4.2).

### 4.1   Fine-tuning Pre-trained Generator for Target Images

GANs typically have a generator $G$ and a discriminator $D$. Given an input signal $z \sim \mathcal{N}(0,1)$, a well-trained generator synthesizes an image $G(z)$. In our tasks, we adapt an off-the-shelf GAN generator $G$ that is pre-trained on the ImageNet-2012 dataset in order to generate more images in a target, data-scarce domain. Note that we do not use the discriminator, since adversarial training with a few images is unstable and may lead to model collapse. Formally, we fine-tune $z$ and the generator $G$ such that $G$ generates an image $\mathbf{I}_z$ from an input vector $z$ by minimizing the distance between $G(z)$ and $\mathbf{I}_z$, where the vector $z$ is randomly initialized. Inspired by previous work [2,5,20], we minimize a loss function $\mathcal{L}_G$ with $\mathcal{L}_1$ distance and perceptual loss $\mathcal{L}_{perc}$ with earth mover regularization $\mathcal{L}_{EM}$,

$$\mathcal{L}_G\left(G, \mathbf{I}_z, z\right) = \mathcal{L}_1\left(G(z), \mathbf{I}_z\right) + \lambda_p \mathcal{L}_{perc}\left(G(z), \mathbf{I}_z\right) + \lambda_z \mathcal{L}_{EM}\left(z, r\right), \tag{1}$$

where $\mathcal{L}_{EM}$ is an earth mover distance between $z$ and random noise $r \sim \mathcal{N}(0,1)$ to regularize $z$ to be sampled from a Gaussian, and $\lambda_p$ and $\lambda_z$ are coefficients of each term.

Since only a few training images are available in the target domain, only scale and shift parameters of the batch normalization of $G$ are updated in practice. Specifically, only the $\gamma$ and $\beta$ of each batch normalization layer are updated in each layer,

$$\hat{x} = \frac{x - \mathbb{E}(x)}{\sqrt{\mathrm{Var}(x) + \epsilon}} \qquad h = \gamma \hat{x} + \beta, \tag{2}$$

where $x$ is the input feature from the previous layer, and $\mathbb{E}$ and $\mathrm{Var}$ indicate the mean and variance functions, respectively. Intuitively and in principle, updating $\gamma$ and $\beta$ only is equivalent to adjusting the activation of each neuron in a layer. Once updated, the $G(z)$ would be synthesized to reconstruct the image $\mathbf{I}_z$. Empirically, a small random perturbation $\epsilon$ is added to $z$ as $G(z + \epsilon)$. Examples of $\mathbf{I}_z$, $G(z)$ and $G(z + \epsilon)$ are illustrated in in Fig. 1 (a), (e), and (f), respectively.

## 4.2 Meta Reinforced Synthetic Data for Few-shot Learning

We propose a meta-learning method to add synthetic data to the originals.

**One-shot Learning.** One-shot classification is a meta-learning problem that divide a dataset into two sets: meta-training (or base) set and meta-testing (or novel) set. The classes in the base set and the novel sets are disjoint. In other words, we have $\mathcal{D}_{base} = \{(\mathbf{I}_i, y_i), y_i \in \mathcal{C}_{base}\}$ and $\mathcal{D}_{novel} = \{(\mathbf{I}_i, y_i), y_i \in \mathcal{C}_{novel}\}$ where $\mathcal{C}_{base} \cup \mathcal{C}_{novel} = \emptyset$. The task is to train a classifier on $\mathcal{D}_{base}$ that can quickly generalize to unseen classes in $\mathcal{C}_{novel}$ with one or few examples. To do this, a meta-learning algorithm performs meta-training by sampling many one-shot tasks from $\mathcal{D}_{base}$, and is evaluated by sampling many similar tasks from $\mathcal{D}_{novel}$. Each sampled task (called an episode) is an $n$-way-$m$-shot classification problem with $q$ queries, meaning that we sample $n$ classes with $m$ training and $q$ test examples for each class. In other words, an episode has a support (or training) set $S$ and a query (or test) set $Q$, where $|S| = n \times m$ and $|Q| = n \times q$. One-shot learning means $m = 1$. The notation $S_c$ means the support examples only belong to the class $c$, so $|S_c| = m$.

**Meta Image Reinforcing Network (MetaIRNet).** We propose a Meta Image Reinforcing Network (MetaIRNet), which not only learns a few-shot classifier, but also learns to reinforce generated images by combining real and generated images. MetaIRNet is composed of two modules: an image fusion network $F$, and a one-shot classification network $C$.

*The Image Fusion Network $F$* combines a real image $\mathbf{I}$ and a corresponding generated image $\mathbf{I}_g$ into a new image $\mathbf{I}_{syn} = F(\mathbf{I}, \mathbf{I}_g)$ that is beneficial for training a one-shot classifier. Among the many possible ways to synthesize images, we were inspired by a block augmentation method [6] and use grid-based linear combination. As shown in Figure 1(g), we divide the images into a $3 \times 3$ grid and linearly combine the cells with the weights $\mathbf{w}$ produced by a CNN conditioned on the two images. That is,

$$\mathbf{I}_{syn} = \mathbf{w} \odot \mathbf{I} + (1 - \mathbf{w}) \odot \mathbf{I}_g \tag{3}$$

where $\odot$ is element-wise multiplication, and $\mathbf{w}$ is resized to the image size keeping the block structure. The CNN to produce $\mathbf{w}$ extracts the feature vectors of $\mathbf{I}$ and $\mathbf{I}_g$, concatenates them, and uses a fully-connected layer to produce a weight corresponding to each of the nine cells in the $3 \times 3$ grid. Finally, for each real image $\mathbf{I}^i$, we generate $n_{aug}$ images, producing $n_{aug}$ synthetic images, and assign the same class label $y_i$ to each synthesized image $\mathbf{I}_{syn}^{i,j}$ to obtain an augmented support set,

$$\tilde{S} = \left\{ \left(\mathbf{I}^i, y^i\right), \left\{ \left(\mathbf{I}_{syn}^{i,j}, y^i\right)\right\}_{j=1}^{n_{aug}} \right\}_{i=1}^{n \times m}. \tag{4}$$

The *One-Shot Classification Network $C$* maps an input image $\mathbf{I}$ into feature maps $C(\mathbf{I})$, and performs one-shot classification. Although any one-shot classifier can be used, we choose the non-parametric prototype classifier of Snell *et al.* [28] due to its superior performance and simplicity. During each episode, given the sampled $S$ and $Q$, the image fusion network produces an augmented support set $\tilde{S}$. This classifier computes the prototype vector $\mathbf{p}_c$ for each class $c$ in $\tilde{S}$ as an average feature vector,

$$\mathbf{p}_c = \frac{1}{|\tilde{S}_c|} \sum_{(\mathbf{I}_i, y_i) \in \tilde{S}_c} C(\mathbf{I}_i). \tag{5}$$

For a query image $\mathbf{I}_i \in Q$, the probability of belonging to a class $c$ is estimated as,

$$P(y_i = c | \mathbf{I}_i) = \frac{\exp(-\|C(\mathbf{I}_i) - \mathbf{p}_c\|)}{\sum_{k=1}^{n} \exp(-\|C(\mathbf{I}_i) - \mathbf{p}_k\|)} \tag{6}$$

where $\| \cdot \|$ is the Euclidean distance. Then, for a query image, the class with the highest probability becomes the final prediction of the one-shot classifier.

**Training**    In the meta-training phase, we jointly train $F$ and $C$ in an end-to-end manner, minimizing a cross-entropy loss function,

$$\min_{\theta_F, \theta_C} \frac{1}{|Q|} \sum_{(\mathbf{I}_i, y_i) \in Q} -\log P\left(y_i \mid \mathbf{I}_i\right),\qquad(7)$$

where $\theta_F$ and $\theta_C$ are the learnable parameters of $F$ and $C$.

## 5  Experiments

To investigate the effectiveness of our approach, we perform 1-shot-5-way classification following the meta-learning experimental setup described in Sec. 4.2. We perform 1000 episodes in meta-testing, with 16 query images per class per episode, and report average classification accuracy and 95% confidence intervals. We use the fine-grained classification dataset of Caltech UCSD Birds (CUB) [32] for our main experiments, and another fine-grained dataset of North American Birds (NAB) [30] for secondary experiments. CUB has 11,788 images with 200 classes, and NAB has 48,527 images with 555 classes.

### 5.1  Implementation Details

While our fine-tuning method introduced in Sec. 4.1 can generate images for each step in meta-training and meta-testing, it takes around 20 seconds per image, so we apply the generation method ahead of time to make our experiments more efficient. We use a BigGAN pre-trained on ImageNet, using the publicly-available weights. We set $\lambda_p = 0.1$ and $\lambda_z = 0.1$, and perform 500 gradient descent updates with the Adam [18] optimizer with learning rate 0.01 for $z$ and 0.0005 for the fully connected layers, to produce scale and shift parameters of the batch normalization layers. We manually chose these hyper-parameters by trying random values from 0.1 to 0.0001 and visually checking the quality of a few generated images. We only train once for each image, generate 10 random images by perturbing $z$, and randomly use one of them for each episode ($n_{aug} = 1$). For image classification, we use ResNet18 [16] pre-trained on ImageNet for the two CNNs in $F$ and one in $C$. We train $F$ and $C$ with Adam with a default learning rate of 0.001. We select the best model based on the validation accuracy, and then compute the final accuracy on the test set. For CUB, we use the same train/val/test split used in previous work [4], and for NAB we randomly split with a proportion of train:val:test = 2:1:1; see supplementary material for details. Further implementation details are available as supplemental source code.[2]

### 5.2  Comparative and Ablative Study on CUB dataset

**Baselines.**    We compare our MetaIRNet with three types of baselines. (1) Non-meta learning classifiers: We directly train the same ImageNet pre-trained CNN used in $F$ to classify images in $\mathcal{D}_{base}$, and use it as a feature extractor for $\mathcal{D}_{novel}$. We then use off-the-shelf classifiers **nearest neighbor**, **logistic regression** (one-vs-all classifier), and **softmax regression** (also called multi-class logistic regression). (2) Meta-learning classifiers: We try the meta-learning method of prototypical network (**ProtoNet** [28]). ProtoNet computes an average prototype vector for each class and performs nearest neighbor with the prototypes. We note that our MetaIRNet adapts ProtoNet as a choice of $F$ so this is an ablative version of our model (MetaIRNet without the image fusion module). (3) Data augmentation: Because our MetaIRNet learns data-augmentation as a sub-module, we also compare with three data augmentation strategies, Flip, Gaussian, and FinetuneGAN. **Flip** horizontally flips the images. **Gaussian** adds Gaussian noise with standard deviation 0.01 into the CNN features. **FinetuneGAN** (introduced in Sec. 4.1) generates augmented images by fine-tuning the ImageNet-pretrained BigGAN with each support set. Note that we do these augmentations in the meta-testing stage to increase the support set. For fair comparison, we use ProtoNet as the base classifier of these data augmentation baselines.

**Results.**    As shown in Table 2, our MetaIRNet is superior to all baselines including the meta-learning classifier of ProtoNet (84.13% vs. 81.73%) on the CUB dataset. It is notable that while ProtoNet has worse accuracy when simply using the generated images as data augmentation, our method shows an

Table 2: 5-way-1-shot accuracy (%) on CUB/NAB dataset with ImageNet pre-trained ResNet18

| Method | Data Augmentation | CUB Acc. | NAB Acc. |
|---|---|---|---|
| Nearest Neighbor | - | 79.00 ± 0.62 | 80.58 ± 0.59 |
| Logistic Regression | - | 81.17 ± 0.60 | 82.70 ± 0.57 |
| Softmax Regression | - | 80.77 ± 0.60 | 82.38 ± 0.57 |
| ProtoNet | - | 81.73 ± 0.63 | 87.91 ± 0.52 |
| ProtoNet | FinetuneGAN | 79.40 ± 0.69 | 85.40 ± 0.59 |
| ProtoNet | Flip | 82.66 ± 0.61 | 88.55 ± 0.50 |
| ProtoNet | Gaussian | 81.75 ± 0.63 | 87.90 ± 0.52 |
| MetaIRNet (Ours) | FinetuneGAN | 84.13 ± 0.58 | 89.19 ± 0.51 |
| MetaIRNet (Ours) | FinetuneGAN, Flip | **84.80 ± 0.56** | **89.57 ± 0.49** |

Table 3: 5-way-1-shot accuracy (%) on CUB dataset with Conv4 without ImageNet pre-training

| MetaIRNet | ProtoNet [28] | MatchingNet [31] | MAML [10] | RelationNet [29] |
|---|---|---|---|---|
| **65.86 ± 0.72** | 63.50 ± 0.70 | 61.16 ± 0.89 [4] | 55.92 ± 0.95 [4] | 62.45 ± 0.98 [4] |

accuracy increase from ProtoNet, which is equivalent to MetaIRNet without the image fusion module. This indicates that our image fusion module can effectively complement the original images while removing harmful elements from generated ones.

Interestingly, horizontal flip augmentation yields nearly a 1% accuracy increase for ProtoNet. Because flipping augmentation cannot be learned directly by our method, we conjectured that our method could also benefit from it. The final line of the table shows an additional experiment with our MetaIRNet combined with random flip augmentation, showing an additional accuracy increase from 84.13% to 84.80%. This suggests that our method provides an improvement that is orthogonal to flip augmentation.

**Case Studies.** We show some sample visualizations in Fig. 4. We observe that image generation often works well, but sometimes completely fails. An advantage of our technique is that even in these failure cases, our fused images often maintain some of the object's shape, even if the images themselves do not look realistic. In order to investigate the quality of generated images in more detail, we randomly pick two classes, sample 100 images for each class, and a show t-SNE visualization of real images (●), generated images (▲), and augmented fused images (+) in Fig. 3, with classes shown in red and blue. It is reasonable that the generated images are closer to the real ones, because our loss function (equation 1) encourages this to be so. Interestingly, perhaps due to artifacts of $3 \times 3$ patches, the fused images are distinctive from the real/generated images, extending the decision boundary.

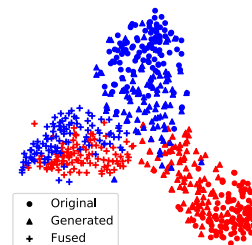

Figure 3: t-SNE plot

**Comparing with state-of-the-art meta-learning classifiers.** It is a convention in the machine learning community to compare any new technique with the performance of many state-of-the-art methods reported in the literature. This is somewhat difficult for us to do fairly, however: we use ImageNet-pre-trained features as a starting point (which is a natural design decision considering that our focus is how to use ImageNet pre-trained generators for improving fine-grained one-shot classification), but much of the one/few-shot learning literature focuses on algorithmic improvements and thus trains from scratch (often with non-fine-grained datasets). The Delta Encoder [25], which uses the idea of learning data augmentation in the feature space, reports 82.2% on one-shot classification on the CUB dataset with ImageNet-pre-trained features, but this is an average of only 10 episodes.

To provide more stable comparison, we cite a benchmark study [4] reporting accuracy of other meta-learners [10, 29, 31] on the CUB dataset with 600 episodes. To compare with these scores, we experimented with our MetaIRNet and the ProtoNet baseline using the same four-layered CNN. As shown in Table 3, our MetaIRNet performs better than the other methods with more than 2%

| Original | Generated | Fused | Weight | Original | Generated | Fused | Weight |
|----------|-----------|-------|--------|----------|-----------|-------|--------|

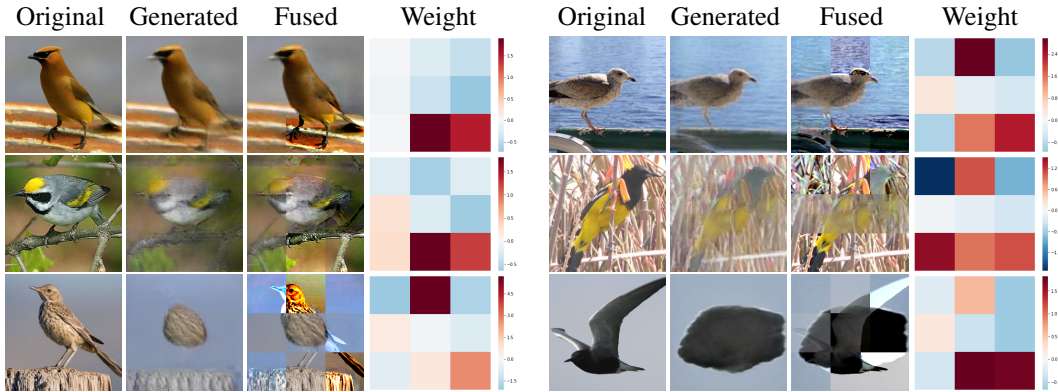

Figure 4: Samples of original image, generated image, fused image, and mixing weight **w**. Higher weight (red) means more original image used, and lower weight (blue) means more generated image used. We show three types of samples based on the quality of generated images: very good (top row), relatively good (middle row), and very bad or broken (last row).

absolute improvement. We note that this comparison is not totally fair because we use images generated from a generator pre-trained from ImageNet. However, our contribution is not to establish a new state-of-the-art score but to present the idea of transferring an ImageNet pre-trained GAN for improving one shot classifiers, so we believe this comparison is still informative.

### 5.3 Results on NAB Dataset

We also performed similar experiments on the NAB dataset, which is more than four times larger than CUB, and the results are shown in the last column of Table 2. We observe similar results as CUB, and that our method improves classification accuracy from a ProtoNet baseline (89.19% vs. 87.91%).

## 6 Conclusion

We introduce an effective way to employ an ImageNet-pre-trained image generator for the purpose of improving fine-grained one-shot classification when data is scarce. As a way to fine-tune the pre-trained generator, our pilot study finds that adjusting only scale and shift parameters in batch normalization can produce a visually realistic images. This technique works with a single image, making the method less dependent on the number of available images. Furthermore, although naively adding the generated images into the training set does not improve performance, we show that it can improve performance if we mix generated with original images to create hybrid training exemplars. In order to learn the parameters of this mixing, we adapt a meta-learning framework. We implement this idea and demonstrate a consistent and significant improvement over several classifiers on two fine-grained benchmark datasets.

### Acknowledgments

We would like to thank Yi Li for drawing Figure 2, and Minjun Li and Atsuhiro Noguchi for helpful discussions. Part of this work was done while Satoshi Tsutsui was an intern at Fudan University. Yanwei Fu was supported in part by the NSFC project (#61572138), and Science and Technology Commission of Shanghai Municipality Project (#19511120700). David Crandall was supported in part by the National Science Foundation (CAREER IIS-1253549), and the Indiana University Office of the Vice Provost for Research, the College of Arts and Sciences, and the Luddy School of Informatics, Computing, and Engineering through the Emerging Areas of Research Project "Learning: Brains, Machines, and Children." Yanwei Fu is the corresponding author.

## Footnotes

[2]http://vision.soic.indiana.edu/metairnet/

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
