[Supplementary Material · supp.pdf]

# 7 Supplementary

## 7.1 Five-shot Experiments

Although our paper focuses on one-shot learning, we also try a five-shot scenario. We use the ImageNet-pretrained ResNet18 [16] as a backbone. We also try the four layer-CNN (Conv-4) without ImageNet pretraining in order to compare with other reported scores in a benchmark study [4]. The results are summarized in Table 4. When we use ImageNet pretrained ResNet, our method is slightly (92.66% v.s 92.83%) better than the ProtoNet. Given that non-meta-learning linear classifiers (softmax regression and logistic regression) can achieve more than 92% accuracy, we believe that ImageNet features are already strong enough when used with five examples per class.

Table 4: 5-way-5-shot Accuracy (%) on CUB dataset.

| Method | Base Network | Initialization | Accuracy |
|---|---|---|---|
| Nearest neighbor | ResNet18 | ImageNet | $89.44 \pm 0.36$ |
| Softmax regression | ResNet18 | ImageNet | $92.28 \pm 0.30$ |
| Logistic regression | ResNet18 | ImageNet | $92.34 \pm 0.30$ |
| ProtoNet [28] | ResNet18 | ImageNet | $92.97 \pm 0.31$ |
| MetaIRNet (Ours) | ResNet18 | ImageNet | $\mathbf{93.09 \pm 0.30}$ |
| MAML [10] | Conv-4 | Random | $72.09 \pm 0.76$ [4] |
| MatchingNet [31] | Conv-4 | Random | $72.86 \pm 0.76$ [4] |
| RelationNet [29] | Conv-4 | Random | $76.11 \pm 0.69$ [4] |
| ProtoNet [28] | Conv-4 | Random | $80.75 \pm 0.46$ |
| MetaIRNet (Ours) | Conv-4 | Random | $\mathbf{81.16 \pm 0.47}$ |

## 7.2 An Implementation Detail: Class label input of BigGAN

Part of the noise $z$ used in BigGAN is class conditional, and we did not explicitly discuss this part in the main paper, so here we provide details. We optimize the class conditional embedding and regard it as part of the input noise. Generally speaking, a conditional GAN uses input noise conditioned on the label of the image to generate. BigGAN also follows this approach, but our fine-tuning technique uses a single image to train. In other words, we only have a single class label and can then optimize the class embedding as part of the input noise.

## 7.3 More Experiments

**Increasing the number of generated examples.** It is interesting to know if our method can benefit by increasing the number of generated examples. We try $n_{aug} = 1, 2, 3, 5, 10$ on CUB and obtain accuracies $84.13 \pm 0.60$, $83.45 \pm 0.60$, $80.99 \pm 0.62$, $81.21 \pm 0.62$, and $80.49 \pm 0.69$, respectively. Too many augmented images seems to bias the classifier. We conclude that the performance gain is marginal or even harmful when increasing $n_{aug}$.

**Mixup baseline.** Mixup [36] uses random $1 \times 1$ weights to mix two images, which can be viewed as a much simpler version of our method to mix a real and generated image pairs. We test this baseline and obtain one-shot accuracy of $82.24 \pm 0.59$ and $88.33 \pm 0.53$ on CUB and NAB, respectively. These results are higher than baselines but still lower than ours.

**Image deformation baseline.** Image deformation net [7] also uses similar $3 \times 3$ patch based data augmentation learning. The key difference is that while that method augments support image by fusing with external real images called a gallery set, our model fuses with images synthesized by GANs. Further, to adapt a generic pretrained GAN to a new domain, we introduce a technique of optimizing only the noise z and BatchNorm parameters rather than the full generator, which is not explored by deformation net [7]. We try this baseline by using a gallery set of random images sampled from the meta-training set, and obtain 1-shot-5-way accuracies of $82.84 \pm 0.62$ on CUB and $88.42 \pm 0.59$ on NAB, which is higher than the baselines but not as high as ours.

**Training from scratch or end-to-end.**    It is an interesting direction to train the generator end-to-end and without ImageNet pretraining. Theoretically, we can do end-to-end training of all components, but in practice we are limited by our GPU memory, which is not large enough to hold both our model and BigGAN. In order to simulate the end-to-end and scratch training, we introduce two constraints. 1) We simplified BigGAN with one-quarter the number of channels and train from scratch so that we train the generator with a relatively small meta-training set. 2) We do not propagate the gradient from classifier to the generator so that we do not have to put both models onto GPU. We apply our approach with a four-layer CNN backbone with random initialization and achieved an 1-shot-5-way accuracy of $63.77 \pm 0.71$ on CUB.

**Experiment on Mini-ImageNet**    . Although our method is designed for fine-grained recognition, it is interesting to apply this to course-grained recognition. Because the public BigGAN model was trained on images including the meta-testing set of ImageNet, we cannot use it as-is. Hence we train the simplified generator (see above paragraph) from scratch using the meta-training set only. Using a backbone of ResNet18, the 1-shot-5-way accuracy on Mini-ImageNet is $53.97 \pm 0.63$ and $55.01 \pm 0.62$ for ProtoNet and MetaIRNet, respectively.

## 7.4    Dataset Split for NAB

We used the following dataset split for the North American Bird (NAB) [30] dataset.

**Label IDs used for training set**  295, 297, 299, 314, 316, 318, 320, 322, 324, 326, 328, 330, 332, 334, 336, 338, 340, 342, 344, 346, 348, 350, 352, 354, 356, 358, 360, 362, 364, 366, 368, 370, 372, 374, 376, 378, 380, 382, 393, 395, 397, 399, 401, 446, 448, 450, 452, 454, 456, 458, 460, 462, 464, 466, 468, 470, 472, 474, 476, 478, 480, 482, 484, 486, 488, 490, 492, 494, 496, 498, 500, 502, 504, 506, 508, 510, 512, 514, 516, 518, 520, 522, 524, 526, 528, 530, 532, 534, 536, 538, 540, 542, 544, 546, 548, 550, 552, 554, 556, 558, 560, 599, 601, 603, 605, 607, 609, 611, 613, 615, 617, 619, 621, 623, 625, 627, 629, 631, 633, 635, 637, 639, 641, 643, 645, 647, 649, 651, 653, 655, 657, 659, 661, 663, 665, 667, 669, 671, 673, 675, 677, 679, 681, 697, 699, 746, 748, 750, 752, 754, 756, 758, 760, 762, 764, 766, 768, 770, 772, 774, 776, 778, 780, 782, 784, 786, 788, 790, 792, 794, 796, 798, 800, 802, 804, 806, 808, 810, 812, 814, 816, 818, 820, 822, 824, 826, 828, 830, 832, 834, 836, 838, 840, 842, 844, 846, 848, 850, 852, 854, 856, 858, 860, 862, 864, 866, 868, 870, 872, 874, 876, 878, 880, 882, 884, 886, 888, 890, 892, 894, 896, 898, 900, 902, 904, 906, 908, 910, 912, 914, 916, 918, 920, 922, 924, 926, 928, 930, 932, 934, 936, 938, 940, 942, 944, 946, 948, 950, 952, 954, 956, 958, 960, 962, 964, 966, 968, 970, 972, 974, 976, 978, 980, 982, 984, 986, 988, 990, 992, 994, 996, 998, 1000, 1002, 1004, 1006, 1008, 1010

**Label IDs used for validation set**  298, 315, 319, 323, 327, 331, 335, 339, 343, 347, 351, 355, 359, 363, 367, 371, 375, 379, 392, 396, 400, 447, 451, 455, 459, 463, 467, 471, 475, 479, 483, 487, 491, 495, 499, 503, 507, 511, 515, 519, 523, 527, 531, 535, 539, 543, 547, 551, 555, 559, 600, 604, 608, 612, 616, 620, 624, 628, 632, 636, 640, 644, 648, 652, 656, 660, 664, 668, 672, 676, 680, 698, 747, 751, 755, 759, 763, 767, 771, 775, 779, 783, 787, 791, 795, 799, 803, 807, 811, 815, 819, 823, 827, 831, 835, 839, 843, 847, 851, 855, 859, 863, 867, 871, 875, 879, 883, 887, 891, 895, 899, 903, 907, 911, 915, 919, 923, 927, 931, 935, 939, 943, 947, 951, 955, 959, 963, 967, 971, 975, 979, 983, 987, 991, 995, 999, 1003, 1007

**Label IDs used for test set**  296, 313, 317, 321, 325, 329, 333, 337, 341, 345, 349, 353, 357, 361, 365, 369, 373, 377, 381, 394, 398, 402, 449, 453, 457, 461, 465, 469, 473, 477, 481, 485, 489, 493, 497, 501, 505, 509, 513, 517, 521, 525, 529, 533, 537, 541, 545, 549, 553, 557, 561, 602, 606, 610, 614, 618, 622, 626, 630, 634, 638, 642, 646, 650, 654, 658, 662, 666, 670, 674, 678, 696, 700, 749, 753, 757, 761, 765, 769, 773, 777, 781, 785, 789, 793, 797, 801, 805, 809, 813, 817, 821, 825, 829, 833, 837, 841, 845, 849, 853, 857, 861, 865, 869, 873, 877, 881, 885, 889, 893, 897, 901, 905, 909, 913, 917, 921, 925, 929, 933, 937, 941, 945, 949, 953, 957, 961, 965, 969, 973, 977, 981, 985, 989, 993, 997, 1001, 1005, 1009