[Reviews · NeurIPS 2019]

Reviewer 1



The results seem to improve over ProtoNet which is basically the only baseline used in the paper despite mentioning lots of data augmentation papers in related work. However, I couldn't quickly equally comparable works using ImageNet data to do meta-learning on CUB (although I am not working in this field). Reproducibility looks okay barring the fact that BigGAN is a conditional GAN and requires class label. Strictly speaking it is not mandatory if the goal is to find BN parameters anyway (which depend on class label in BigGAN), but I imagine the initialization matters. I'd appreciate clarifications about this aspect of the implementation. The writing is fairly clear, although section 4 contains quite verbose notation. The idea is not particularly original, many tried using GANs for data augmentation. But this implementation does work, even though the improvement is not ground-breaking.

Reviewer 2



Paper Strengths: The authors tackle an important and challenging problem of few-shot fine-grained classification. The proposed approach is simple. Experimental evaluations demonstrate the effect by modifying the GAN generated images by combining them with real images at the patch level through meta-learning. Paper Weaknesses: 1) The proposed approach in this paper can be viewed as a straightforward combination of an off-the-shelf GAN and [7] that learns to linearly fuse two images for data augmentation. The novelty of the proposed approach is somewhat limited. In addition, the connection with [7] is not fully discussed. 2) Since [7] is directly relevant to the proposed approach, it would be more convicting to show the experimental comparison with [7]. 3) The technique in the proposed approach seems not restricted to fine-grained recognition. It would be interesting to evaluate the approach on standard generic image few-shot benchmark such as miniImageNet. 4) How is the performance if we do not use a pre-trained GAN generator but train the generator also in an end-to-end manner? 5) How is the performance with respect to the number of generated examples? 6) Why does the proposed approach work for few-shot learning? It looks like that the GAN image is similar to the original real image. Hence, their combination does not introduce diverse examples beyond the given real images.

Reviewer 3



Originality: 7 / 10 This is a novel paper that is well motivated and executed. Admittedly, all of its components are not novel alone -- grid linear mixture for image augmentation [6], meta-learned generator [35], episodical procedure, and standard few-shot classifiers. The proposed pipeline itself is new and does provide insight that end-to-end image-augmentation is feasible with a strong generator initialization. And also, finetuning a GAN towards certain modalities (or observations) are not informatively studied before. Figure 1 and its experiments could serve as a good reference to researchers who want to study image augmentation. ---------- Quality: 8 / 10 The experiments, as well as the pilot study, are in great shapes and considerations. The shown performance gains are consistent and non-trivial compared to its previous works. One thing to note is that the baselines are rather old -- the most contemporary model in the comparison table is RelationNet [29] from CVPR 2018. I believe there are still a lot of missing numbers out there. And what will make this paper even better are experiments on ImageNet itself to see how it will scale to large benchmark. But I am not exactly sure if BigGAN has seen the testing set during its training. ---------- Clarity: 8 / 10 The writing is good and it was a great pleasure reading it. Notations are consistent and there are very few typos that do not actually interrupt the flow. It is such a pity that authors have not included the code, but I would strongly suggest to opensource as early as possible since people will be following it. ---------- Significance: 8 / 10 It would be better if there's any intuition or even empirical study on the correlation between the fusing weight and the real/synthesized image pairs. ---------- Other comments: (1) How would your augmented fused image embed on feature space? And how will it affect the final decision of the classification head? Since you are primarily focused on the one-shot learning task, it will be informative and also doable for such illustration. (2) L180, typo, inconsistent $\mathbf{I_g}$, change it to $\mathbf{I}_g$. (3) L114, logistic regression, is it a one-versus-all classifier for each category, and then select the most salient probability from all candidates? There should be more text for description. (4) L141, $\mathbf{I}_z$, missing definition, this is the first time you use this notation without any reference.

[Author Response · NeurIPS 2019]

We thank the reviewers for their constructive feedback. We are pleased that they appreciated our "novel paper that is well motivated and executed" [R4], that "tackles an important and challenging problem of few-shot fine-grained classification" [R3], and that "will draw impact to both rigorous communities of few-shot and fine-grain recognition" [R4]. "The experiments, as well as the pilot study, are in great shape" [R4]. Our "framework works well on reasonably difficult datasets" [R1], and "can be useful for the future research" [R1]. Additionally, we would like to highlight a key contribution of our work: while GAN-generated images have not generally been useful for training image recognition models [26], we show how to effectively use such generated images for one-shot learning for fine-grained recognition.

**R1: Class label input of BigGAN**: We optimize the class conditional embedding and regard it as part of the input noise. Generally speaking, a conditional GAN uses input noise conditioned on the label of the image to generate. BigGAN also follows this approach, but our fine-tuning technique uses a single image to train. In other words, we only have a single class label and can then optimize the class embedding as part of the input noise. We will clarify this point.

**R1: Compare with Mixup**: As suggested, we ran Mixup as a baseline and obtained one-shot accuracy of $82.65 \pm 0.59$ and $88.12 \pm 0.52$ on CUB and NAB, respectively. These results are higher than baselines but still lower than ours.

**R3: Compare with [7]**: Data augmentation is also used in [7], published at CVPR 2019. The key difference is that while they augment support image by fusing with external real images from a gallery set, our model fuses with images synthesized by GANs, and it is non-trivial to make this change. Further, to adapt a generic pretrained GAN to a new domain, we introduce a technique of optimizing only z and BatchNorm parameters rather than the full generator, which is a novel aspect compared to [7]. Nevertheless, as suggested, we implemented their approach and obtained accuracies of $82.84 \pm 0.62$ on CUB and $88.42 \pm 0.59$ on NAB, which is higher than the baselines but not as high as ours.

**R3: Train end-to-end from scratch**: Theoretically, we can do end-to-end training of all components, but in practice we are limited by our GPU memory, which is not large enough to hold both our model and BigGAN. However, to validate this point, we have added another experiment. We simplified BigGAN with one-quarter the number of channels and applied the rest of our approach with a backbone of four-layer CNN with random initialization. This model was trained end-to-end on CUB meta-training dataset, and achieved an accuracy of $63.77 \pm 0.71$, which is still higher than the baselines.

**R3: Performance w.r.t. the number of synthesized images**: As suggested, we increased $n_{aug} = 1, 2, 3, 5, 10$ on CUB and achieved accuracies $83.51 \pm 0.60, 83.65 \pm 0.60, 81.79 \pm 0.62, 80.79 \pm 0.62, 80.34 \pm 0.63$, and $79.75 \pm 0.69$, respectively. Too many augmented images seem to bias the classifier. We conclude that the performance gain is marginal or even harmful when increasing $n_{aug}$.

**R3: Why our method works**: We do not have firm theoretical explanations for why our method works beyond empirical evidence, but we would like to share some insights that guide us. The data distribution of GAN-generated images is biased towards frequent patterns (or modes) of the original image distribution [26], and may not help train image recognition. Our model helps diversify the data distribution by injecting artifacts of $3 \times 3$ patches, and thus can potentially help recognition. This point is supported by the t-SNE visualization below. R3 is correct that the fused images do not look very different from the originals for humans, but this might not be the case for CNNs; for example, adversarial noise is typically imperceptible to humans but dramatically changes CNN representations.

**R4: More visualization and studies on learned augmentation**: We will add more visualizations similar to Figure 3 in the supplemental material. We are happy to perform other suggestions for additional experiments.

**R4: Fused image on feature space**: As plotted in the right, we randomly pick two classes shown in red and blue, sample 100 images for each class, and apply t-SNE visualization of real images (●), generated images (▲), and augmented fused images (+). It is reasonable that the generated images are closer to the real ones, because our loss function (equation 1) encourages this to be so. Interestingly, perhaps due to artifacts of $3 \times 3$ patches, the fused images are distinctive from the real/generated images in the embedding space, extending the decision boundary.

**R3, R4: Evaluation on ImageNet**: Thank you for the suggestion, but we did not have enough computation resources to do this within the author response time period. Our goal was fine-grained recognition, which is why we did not perform ImageNet experiment originally. Nevertheless, we will include results on ImageNet in the camera ready. The public BigGAN model was trained on images including the meta-testing set of ImageNet, so we will have to train the model from scratch using meta-training set only.

**Others**: Thanks for the suggestions and corrections. We use one-vs-all classifier for logistic regression and will clarify.

[Meta-Review · NeurIPS 2019]

This paper was reviewed by three experts in the field and received 677 recommendations. The reviewers found the proposed approach straightforward and incremental in terms of novelty, but all appreciated its effectiveness demonstrated in the experiments. R1 liked the particular implementation of using GAN for data augmentation work. R3 additionally liked the investigation into BigGAN-based data augmentation. Based on the reviewers' feedback, the decision is to recommend the paper for acceptance to NeurIPS 2019. The reviewers did raise some valuable concerns, for example the additional experiments suggested by R2. These questions should be addressed in the final camera-ready version of the paper. The authors are encouraged to make the necessary changes to the best of their ability. We congratulate the authors on the acceptance of their paper!